# Application of Biotransformation-Guided Purification in Chinese Medicine: An Example to Produce Butin from Licorice

**Jiumn-Yih Wu** [1,†] **Hsiou-Yu Ding** [2,†] **, Tzi-Yuan Wang** [3,†] **, Cheng-Zhi Cai** [4] **and Te-Sheng Chang** [4,*]

1 Department of Food Science, National Quemoy University, Kinmen County 89250, Taiwan; wujy@nqu.edu.tw
2 Department of Cosmetic Science, Chia Nan University of Pharmacy and Science, No. 60 Erh-Jen Rd., Sec. 1, Jen-Te District, Tainan 71710, Taiwan; ding8896@gmail.com
3 Biodiversity Research Center, Academia Sinica, Taipei 11529, Taiwan; tziyuan@gmail.com
4 Department of Biological Sciences and Technology, National University of Tainan, Tainan 70005, Taiwan; i94663075@mailst.cjcu.edu.tw
* Correspondence: mozyme2001@gmail.com; Tel./Fax: +886-6-2602137
† These authors contributed equally to this work.

**Abstract:** Natural compounds are considered treasures in biotechnology; however, in the past, the process of discovering bioactive compounds is time consuming, and the purification and validation of the biofunctions and biochemistry of compounds isolated from a medicinal herb are tedious tasks. In this study, we developed an economical process called biotransformation-guided purification (BGP), which we applied to analyze licorice, a traditional Chinese medicine widely used in many therapies. This medicinal herb contains various flavonoids and triterpenoids and, thus, is a suitable material used to assess the ability of BGP to identify and produce bioactive compounds. In the BGP process, the ethyl acetate extract of a commercial licorice medicine was partially purified into three fractions by Sephadex LH-20 chromatography, and *Bacillus megaterium* tyrosinase (*Bm*TYR) was used to catalyze the biotransformation of the extract from each fraction. One of the products produced via *Bm*TYR-driven biotransformation was purified from the biotransformation-positive extract using preparative C-18 high-performance liquid chromatography, and it was identified as butin (3′-hydroxyliquiritigenin) through nucleic magnetic resonance and mass spectral analyses. Butin was produced from liquiritigenin through *Bm*TYR-catalyzed hydroxylation, with commercial liquiritigenin as the biotransformation precursor. The proposed alternative approach quickly identified and isolated the biotransformed butin from licorice. Moreover, butin demonstrated an antioxidant activity that is stronger by over 100-fold compared with that of its precursor (liquiritigenin). This study showed that the economical BGP process could quickly obtain and validate bioactive molecules from crude extracts of medicinal herbs.

**Keywords:** *Glycyrrhiza*; licorice; tyrosinase; biotransformation; hydroxylation; antioxidant

## 1. Introduction

The production of bioactive compounds is important for drug development. In obtaining bioactive compounds, an alternative strategy to chemical synthesis and/or direct isolation of natural compounds is biotransformation. Biotransformation reactions include hydroxylation, dehydrogenation, lactone formation, methylation, and (de)glycosylation [1–3]. Generally, precursors isolated from natural compounds can be biotransformed into new bioactive compounds by microorganisms or by pure enzymes. Some bioactive compounds with modified functional groups may demonstrate increased bioactivity compared with their precursors.

Among the biotransformation reactions, hydroxylation has been attracting considerable attention from scientists. Additional hydroxyl group(s) not only strengthen(s) the original bioactivities of some precursors, but also impart(s) novel bioactivity to them [4]. Kim et al. have developed an efficient hydroxylation system for flavonoids using *Bacillus*

*megatherium* tyrosinase (*Bm*TYR) in the presence of ascorbate and borate [5]. In our previous study, we also applied tyrosinase to hydroxylate soybean isoflavone, daidzin, and genistin to produce hydroxylated isoflavones [6]. The above studies have revealed that *Bm*TYR is a promiscuous biocatalyst for flavonoid hydroxylation.

Licorice, which contains numerous flavonoids and triterpenoids, is a commonly used traditional Chinese medicine and is a natural sweetening agent. This medicinal herb has been used to treat allergic inflammatory and cardiovascular diseases, and it has been used to relieve coughing and to eliminate phlegm [7,8]. Commercial medicines have been extracted from the roots and rhizomes of *Glycyrrhiza uralensis* Fisch., *G. glabra* L., or *G. inflata* Bat. Modern pharmacological studies have verified that licorice herbs demonstrate hypoglycemic, antimicrobial, antiviral, antioxidant, antidiabetic, antiallergic, anti-inflammatory, and hepatoprotective activities [9,10]. In addition, geographically isolated licorice herbs contain various components with medicinal properties. A recent study has shown that different *Glycyrrhiza* species contain different flavonoids [11–13]. The high diversity of flavonoid content of licorice renders this herb a complex and challenging material to study in terms of biotransformation.

A typical study approach in validating the bioactivities of medicinal herbs is analyzing the functions of pure compounds (precursors). In biotransformation studies, individual pure compounds (precursors) are further analyzed. However, the high cost of these pure precursors limits the conduct of such investigations. The use of herb extracts is another means of conducting studies at a low cost, but this approach entails many challenges due to the presence of various complex compounds in extracts. Therefore, in this study, a biotransformation-guided purification (BGP) process was designed and tested on a commercial licorice medicine to identify the bioactive compounds present in the herb extracts (Figure 1). The herb was first extracted, portioned, and separated into fractions by liquid chromatography. The extract in each fraction was used as the crude precursor to quickly screen the compounds biotransformed by *Bm*TYR. The positively biotransformed precursors and products were subsequently isolated from the fractions and then purified for further functional validation. The BGP process successfully identified one possible precursor compound from licorice herb, and it validated the antioxidant activity of both the precursor and the biotransformed product.

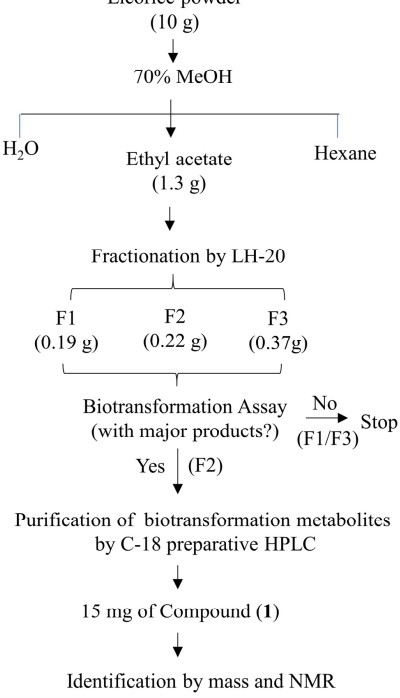

**Figure 1.** Flowchart of the biotransformation-guided purification (BPG) of a commercial licorice herb.

## 2. Results and Discussion

### 2.1. Biotransformation of Licorice Extract by BmTYR

To identify the bioactive flavonoids from the natural extracts, we extracted a commercial licorice herb using methanol, and then we partitioned the extract into $H_2O$–EA–hexane fractionations (Figure 1). The EA extract containing the greatest number of flavonoids was selected for the evaluation of the biotransformation efficiency of *Bm*TYR. The results showed no significant product after biotransformation, which implies that the enzyme could not directly catalyze the hydroxylation of flavonoids in the crude extract (Figure 2). A possible reason is that *Bm*TYR was inhibited by specific flavonoids in the extract [14].

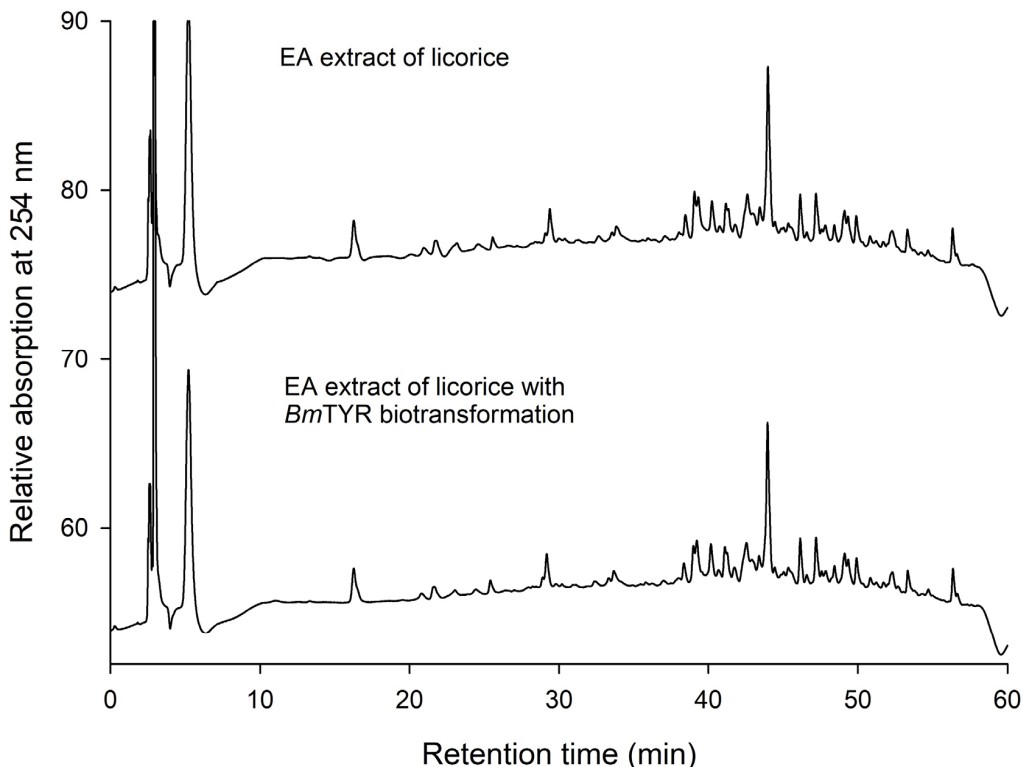

**Figure 2.** High-performance liquid chromatography (HPLC) analysis of the products biotransformed from the ethyl acetate (EA) extract of licorice using *Bm*TYR. The biotransformation mixture containing 108 µg/mL of the purified recombinant *Bm*TYR enzymes, 2 mg/mL of the EA extract, 10 mM ascorbic acid, and 500 mM borate buffer at pH 9 was incubated at 50 °C and 200 rpm-shaking for 1.5 h. At the end of the reaction, one-fifth volume of 1 M HCl and an equal volume of MeOH were added to stop the reaction and was analyzed by HPLC. The HPLC operation procedure was described in Section 3.7.

The above problem could be partially addressed by the proposed BGP process (Figure 1). To separate the inhibitors within the extract, we fractionated the EA extract using Sephadex LH-20 chromatography, obtaining three fractions (F1, F2, and F3) (Figure 1). Each fraction was re-evaluated in terms of *Bm*TYR-catalyzed biotransformation; the biotransformed products were subsequently analyzed by HPLC (Figure 3). Figure 3b shows that the F2 fraction could be biotransformed into a significant product, denoted as compound (**1**). No significant product was identified in F1 and F3 (Figure 3a,c). The biotransformation product from F2 was then further studied.

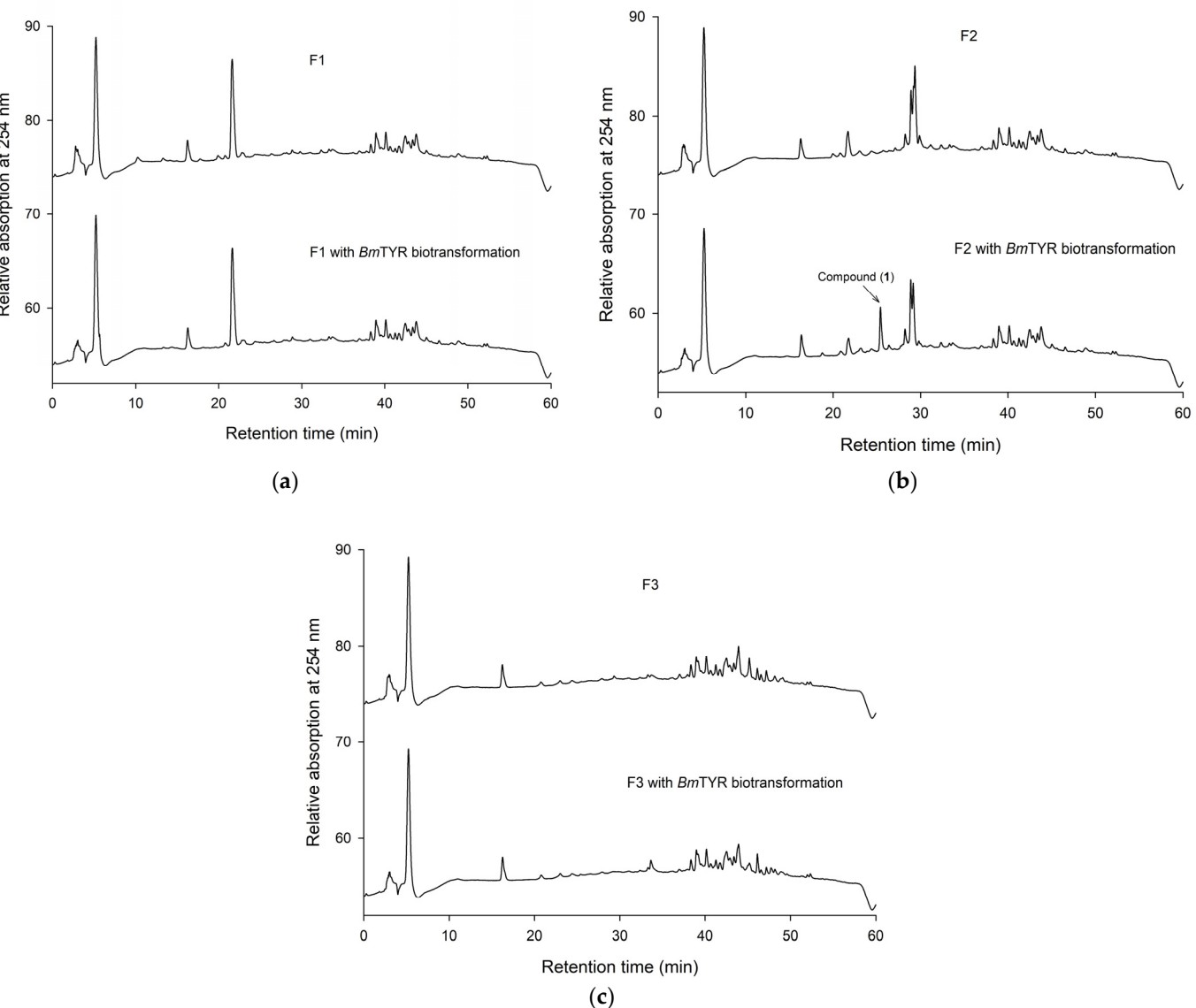

**Figure 3.** High-performance liquid chromatography (HPLC) analysis of the products biotransformed from F1 (**a**), F2 (**b**), and F3 (**c**) extracts using *Bm*TYR. The biotransformation and HPLC conditions were the same as those in Figure 2, except that the enzyme substrate (fraction extracts) is replaced with the indicated fractionation extract.

*2.2. Purification and Identification of the Biotransformation Product from F2*

To resolve the chemical structure of compound (**1**), we scaled up the biotransformation containing the F2 fraction to 100 mL. The *Bm*TYR-biotransformed compound (**1**) was purified by preparative HPLC. The chemical structure of the purified compound (**1**) was then analyzed using mass and nucleic magnetic resonance (NMR) spectral analyses. The molecular formula of compound (**1**) was established as $C_{15}H_{12}O_5$ by the ESI-mass spectrometry (MS) at $m/z$ 271.1 [M-H]$^-$, indicating a molecular weight of 272 (Figure S1). Based on the NMR spectral analysis of compound (**1**), the signals for H-2, H-3a, H-3b, H-5, H-6, H-8, H-2′, H-5′, and H-6′ appeared at $\delta$ 5.38, 2.62, 3.04, 7.63, 6.50, 6.33, 6.88, 6.73, and 6.74 ppm in the $^1$H NMR spectrum, respectively, and the signals for C-2, C-3, C-5, C-6, C-8, C-2′, C-5′, and C-6′ were observed at $\delta$ 79.0, 43.3, 128.4, 110.5, 102.6, 114.3, 115.3, and 117.9 ppm in DMSO-$d_6$ in the $^{13}$C NMR spectrum, respectively, and these findings were confirmed by data in the literature [15]. The full assignments of the $^1$H and $^{13}$C-NMR signals were further indicated by the distortionless enhancement by polarization transfer (DEPT), heteronuclear single quantum coherence (HSQC), heteronuclear multiple bond connectivity (HMBC),

correlation spectroscopy (COSY), and nuclear Overhauser effect spectroscopy (NOESY) spectra, as shown in Table S1 and Figures S2–S8. These data confirmed that compound (**1**) is butin (Figure 4). A study has identified some triterpenoid saponins in the direct biotransformation of licorice [16]. The current study further identified that *Bm*TYR could produce butin from licorice by BGP.

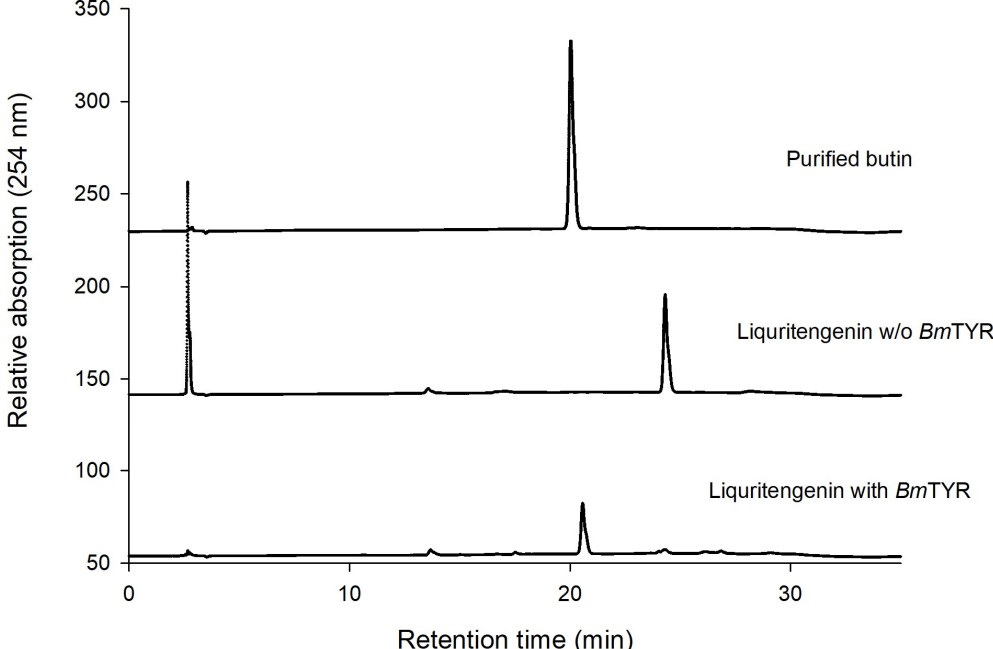

**Figure 4.** Structure of the biotransformation product butin.

### 2.3. Confirming the Biotransformation Process Using Reverse Analysis

The structure of butin is 3′-hydroxyliquiritigenin, a putative compound of licorice that may be hydroxylated by *Bm*TYR. This finding suggested that the putative precursor is liquiritigenin. Liquiritigenin has been reported to be a major flavonoid in licorice herbs, such as *G. uralensis* Fisch., *G. glabra* L., or *G. inflata* Bat [11], although different licorice species have different flavonoid compositions [11–13]. To confirm the biotransformation reaction, we used a commercial liquiritigenin as the pure precursor biotransformed by *Bm*TYR, and the products were analyzed by HPLC. Figure 5 shows that liquiritigenin could be hydroxylated by *Bm*TYR to produce a metabolite, whose retention time in the HPLC analysis was the same as that of the purified butin. Figure 6 illustrates the biotransformation process of liquiritigenin by *Bm*TYR. The BGP process could indeed simplify and identify the enzymatic bioactive compound, and it may be used to identify natural products derived from medicinal herbs.

**Figure 5.** HPLC analysis of the biotransformation products of liquiritigenin obtained using *Bm*TYR. The biotransformation and HPLC conditions were the same as those in Figure 2, except that the enzyme substrate (fraction extracts) is replaced with a liquiritigenin standard.

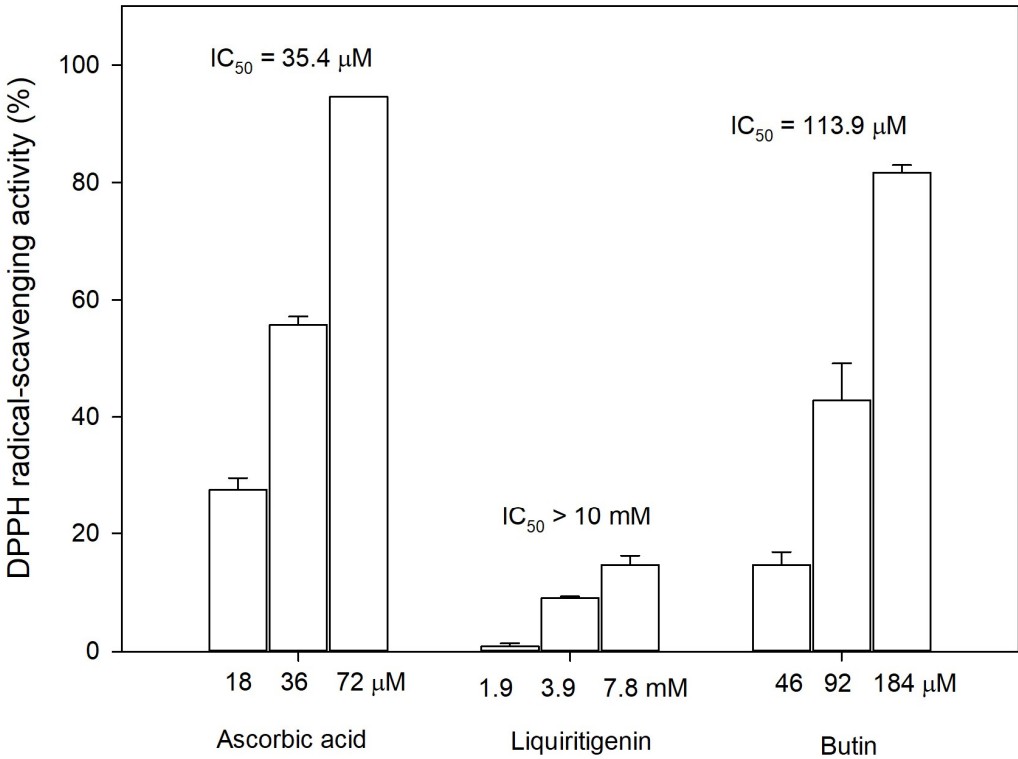

**Figure 6.** The biotransformation of liquiritigenin catalyzed by *Bm*TYR.

## 2.4. Antioxidant Activity of Liquiritigenin and Butin

The *ortho*-dihydroxyl groups on the benzene ring of flavonoid structures have been reported to play an important role in the antioxidant activity of flavonoids [17]. Thus, the antioxidative activities of both butin and its precursor liquiritigenin were determined by the DPPH free radical scavenging assay. The assay showed that the antioxidant activities of butin were over 100-fold stronger than those of liquiritigenin and were comparable to those of ascorbic acid (Figure 7). In addition to the good antioxidant activity, butin also possesses other bioactivities such as skin whitening [18], cardioprotection [19], neuroprotection [20], and cytoprotection [21].

**Figure 7.** 1,1-Diphenyl-2-picrylhydrazine (DPPH) free radical-scavenging activity of liquiritigenin, butin, and ascorbic acid. The determination of the DPPH scavenging activity is described in Section 3.9. The mean (*n* = 3) is shown, and the standard derivative values were represented by error bars. The IC$_{50}$ values represent the concentrations required for 50% DPPH free radical scavenging activity.

In this study, the BGP process was tested on a natural extract (licorice) with three crude extract fractions, and *Bm*TYR was used to confirm a positive biotransformation reaction in one fraction. The results showed that *Bm*TYR could produce a significant amount of biotranformed butin. With the use of the BGP process, valuable precursors found in natural

sources can be studied more easily through enzymatic biotransformation in order to identify bioactive compounds.

Some studies reported that butin could be isolated from natural sources [15,22,23]. The typical purification process of natural products includes (1) crude extraction, (2) partition, and (3) repeated chromatography with different resins. The BGP process introduces an enzymatic biotransformation process into the abovementioned repeated chromatography to isolate bioactive compounds. Thus, one can isolate the biotransformation products and ignore the other unreacted compounds although they can be further purified later. Taking butin as an example, in a typical purification process, 55 mg butin could be purified from 1 kg seeds of *Vernonia anthelmintica* with recovery yield (0.0055%) [15], and 30 mg butin could be purified from 12 kg dried vine stems of *Spatholobus suberectus* with yield (0.00025%) [22], and 12 mg butin could be purified from 11.5 kg roots of *Caragana pruinosa* with yield (0.0001%) [23]. In contrast, through BGP process, 15 mg butin could be purified from 10 g licorice medicine with higher yield (0.15%), which are 27-, 1500-, and 600-folds higher than those from seeds of *Vernonia anthelmintica*, stems of *Spatholobus suberectus*, and roots of *Caragana pruinosa*, respectively. Thus, one of the largest advantages of the BGP process is the ability to purify higher yields of bioactive compound (butin) biotransformed from natural sources (licorice medicine), especially when the natural sources contain high amount of biotransformation precursors (liquiritigenin within licorice medicine) (Figure 1). In addition, a good biotransformation enzyme, such as *Bm*TYR [5,6], is another key issue for a better conversion rate in the BGP process.

## 3. Materials and Methods

### 3.1. Chemicals and Microorganism

*Bacillus megaterium* BCRC 10608 was obtained from the previous study [6]. A commercial licorice medicine (Kan-Tsao herb extract, Sun Ten Pharmaceutical Co., Ltd., Taichung, Taiwan) was purchased from a local pharmacy. Liquiritigenin standard was purchased from Baoji Herbest Bio-Tech (Xi-An, Shaanxi, China). Furthermore, 1,1-Diphenyl-2-picrylhydrazine (DPPH), dimethyl sulfoxide (DMSO), l-dihydroxyphenylalanine (L-DOPA), and ascorbic acid were purchased from Sigma (St. Louis, MO, USA). Recombinant *Bm*TYR was purified from the recombinant *E. coli* BL21 (DE3), which was constructed in this study, as described below. Sephadex LH-20 gel used for the partial purification of licorice was purchased from GE Healthcare (Chicago, IL, USA). The other reagents and solvents used were commercially available.

### 3.2. Construction of Expression Plasmid and Recombinant E. coli

The genomic DNA of *B. megaterium* was isolated using the commercial kit Geno Plus. The *B. megaterium* tyrosinase gene (GenBank protein database accession number KGJ77254) was amplified from the genomic DNA by PCR using the following primers: forward: 5′-cccgaattcgagtaacaagtacagagttagaa-3′; reverse: 5′-cccagatctttatgatgaacgttttgattttc-3′. Restriction enzyme recognition sites (underlined) were designed both in the forward (EcoRI) and reverse (BglII) primers. The amplified tyrosinase gene was subcloned into the pETDuet-1™ vector through the EcoRI and BglII sites to obtain the expression vector pETDuet-*Bm*TYR (Figure S9). The expression vector was subsequently transformed into *E. coli* BL21 (DE3) via electroporation to obtain the recombinant *E. coli*.

### 3.3. Expression and Purification of BmTYR in E. coli

Recombinant *Bm*TYR was prepared according to a previous study [24]. The recombinant *E. coli* harboring the recombinant expression plasmid pETDuet-*Bm*TYR was cultivated in Luria–Bertani medium to an optical density at 560 nm ($OD_{560}$) of 0.6 and then induced with 0.2 mM isopropyl β-D-1-thiogalactopyranoside (IPTG) and 1 mM $CuSO_4$. After further cultivation at 18 °C for 20 h, the cells were broken through sonication using a Branson S-450D Sonifier (Branson Ultrasonic Corp., Danbury, CT, USA). The mixture was subsequently centrifuged and the supernatant containing the recombinant *Bm*TYR fused

with a His-tag in its N-terminal was applied in a $Ni^{2+}$ affinity column. The purified *Bm*TYR showed a single band in SDS-PAGE (Figure S10), and it was stored in a final concentration of 50% glycerol at $-80$ °C before use.

### 3.4. Determination of Tyrosinase Activity

Tyrosinase activity was measured as previously reported [6]. The reaction mixture containing 0.1 mL 2.5 mM l-DOPA (dissolved in PB at pH 6.8) and 108 μg/mL *Bm*TYR was incubated at 25 °C for 2 min. Dopachrome formation ($\varepsilon = 3600$ $M^{-1}$ $cm^{-1}$) in each reaction was monitored using a microplate reader (Sunrise, Tecan, Männedorf, Switzerland) at 475 nm. One unit of *Bm*TYR activity was defined as the amount of enzyme that released 1 μmol of dopachrome per minute under the assay conditions described earlier. The specific L-DOPA oxidation activity of the purified recombinant *Bm*TYR was determined to be 4.84 U/mg.

### 3.5. Biotransformation Using BmTYR

The biotransformation system was operated according to the method reported by Lee et al., with minor modifications [5,6]. The reaction mixture (100 μL) containing 500 mM borate (pH 9.0), 10 mM ascorbic acid, and 2 mg/mL of the tested substrate compound (diluted from a stock of 20 mg/mL in DMSO) and 108 μg/mL of *Bm*TYR was incubated at 50 °C and 200 rpm-shaking for 1.5 h. At the end of the reaction, 20 μL of 1 M HCl and 120 μL of MeOH were added to stop the reaction followed by an analysis using high-performance liquid chromatography (HPLC).

### 3.6. Fractionation of Licorice

The purification process is shown in Figure 1. A commercial licorice powder (10 g) was extracted with 100 mL 70% methanol at 25 °C for 24 h. The mixture was passed through a Watman filter paper, and the filtrate was condensed under a vacuum condition. The condensed pellet was subsequently resuspended in 100 mL of $H_2O$ and n-hexane (1:1). The n-hexane fraction was discarded. Both the $H_2O$ fraction and the insoluble pellet were extracted with 100 mL ethyl acetate (EA). The EA fraction was condensed under a vacuum condition, and an EA extract (1.3 g) was obtained.

The EA extract was resuspended in 20 mL methanol, and the mixture was subsequently separated via LH-20 chromatography (40 i.d. $\times$ 400 mm). The elution was methanol, and its fractions were collected and analyzed by HPLC. Fractions with similar contents were combined, forming three fractions (denoted as F1, F2, and F3), which in turn were condensed under a vacuum condition. Finally, 0.19 g of F1, 0.22 g of F2, and 0.37 g of F3 were obtained.

### 3.7. HPLC Analysis

HPLC was performed on an Agilent 1100 series HPLC system (Santa Clara, CA, USA) equipped with a gradient pump (Waters 600, Waters, Milford, MA, USA). The stationary phase was a C18 column (5 μm, 4.6 i.d. $\times$ 250 mm; Sharpsil H-C18, Sharpsil, Beijing, China), and the mobile phase was 1% acetic acid in water (A) and methanol (B). The elution conditions for the analysis of crude extracts and their biotransformation products were as follows: a linear gradient from 0 min with 30% B to 20 min with 50% B, a linear gradient from 20 min with 50% B to 50 min with 80% B, an isocratic elution from 50 min to 55 min with 80% B, and an isocratic elution from 55 min to 60 min with 30% B. The elution conditions for the analysis of liquiritigenin standard and its biotransformation product were as follows: a linear gradient from 0 min with 40% B to 20 min with 70% B, an isocratic elution from 20 min to 25 min with 70% B, a linear gradient from 25 min with 70% B to 28 min with 40% B, and an isocratic elution from 28 min to 35 min with 40% B. All eluants were eluted at a flow rate of 1 mL/min. The sample volume was 10 μL. The detection condition was set at 254 nm.

### 3.8. Purification and Identification of the Biotransformed Metabolite

The purification process had been previously described [24]. To purify compound (**1**), the biotransformation reaction containing the F2 fraction extract was scaled up to 100 mL (0.5 mL per tube), and the 200-vials reactions were incubated with 200 rpm of shaking at 50 °C for 1.5 h. After reaction, compound (**1**) was purified by a preparative YoungLin HPLC system. The fraction with the compound (**1**) was collected, condensed under a vacuum, and then dehydrated by freeze drying. Finally, 15.0 mg of compound (**1**) was obtained, and the structure of the compound was confirmed through nucleic magnetic resonance (NMR) and mass spectral analyses.

### 3.9. Determination of Antiradical Activity Using DPPH Assay

DPPH assay was performed as previously described, with minor modifications [24]. The tested sample (dissolved in DMSO) was added to the DPPH solution (1 mM in methanol) to a final volume of 0.1 mL. After 15 min of reaction, the absorbance of the reaction mixture was measured at 517 nm using a microplate reader (Sunrise, Tecan, Männedorf, Switzerland). Ascorbic acid (dissolved in DMSO) was used as the positive antioxidant standard. The DPPH free radical scavenging activity was calculated as follows: DPPH free radical scavenging activity = ($OD_{517}$ of the control reaction − $OD_{517}$ of the reaction)/($OD_{517}$ of the control reaction). The $IC_{50}$ value was defined as the concentration of an inhibitor required to scavenge 50% of the initial DPPH free radical under the assay conditions.

## 4. Conclusions

Butin was isolated from licorice crude extract through a BGP process. The flavonoid demonstrated a 100-fold higher DPPH free radical-scavenging activity compared with its precursor (liquiritigenin). This study showed that the BGP process is an alternative means to isolate bioactive molecules from crude extracts of natural sources.

**Supplementary Materials:** The following supporting information can be downloaded at: https://www.mdpi.com/article/10.3390/catal12070718/s1. Table S1. [1]H and [13]C NMR assignments in DMSO-$d_6$ at 700 and 175 MHz for compound (**1**). ($\delta$ in ppm, *J* in Hz). Figure S1. The mass-mass analysis of compound (**1**) at the negative mode. A significant signal at *m/z* 271.1. Figure S2. 1D NMR spectrum ([1]H-NMR, 700 MHz, DMSO-$d_6$) of the compound (**1**). Figure S3. 1D NMR spectrum ([13]C-NMR, 175 MHz, DMSO-$d_6$) of the compound (**1**). Figure S4. 1D NMR spectrum (DEPT-135, 175 MHz, DMSO-$d_6$) of the compound (**1**). Figure S5. 2D NMR spectrum ([1]H-[13]C HSQC, 700 MHz, DMSO-$d_6$) of the compound (**1**). Figure S6. 2D NMR spectrum ([1]H-[13]C HMBC, 700 MHz, DMSO-$d_6$) of the compound (**1**). Figure S7. 2D NMR spectrum ([1]H-[1]H COSY, 700 MHz, DMSO-$d_6$) of the compound (**1**). Figure S8. 2D NMR spectrum ([1]H-[1]H NOESY, 700 MHz, DMSO-$d_6$) of the compound (**1**). Figure S9. The expression plasmid pETDuet-*Bm*TYR. Figure S10. Sodium dodecyl sulfate-polyacrylamide gel electrophoresis (SDS-PAGE) analysis of expressed and purified *Bm*TYR from recombinant *E. coli* haboring pETDuet-*Bm*TYR.

**Author Contributions:** Conceptualization: T.-S.C.; data curation and methodology: C.-Z.C., T.-S.C. and H.-Y.D.; project administration: T.-S.C. and J.-Y.W.; writing—original draft, review, and editing: T.-S.C., T.-Y.W., J.-Y.W. and H.-Y.D. All authors have read and agreed to the published version of the manuscript.

**Funding:** This research was funded by the Ministry of Science and Technology of Taiwan under grant number MOST 110-2221-E-024-002 to T.-S.C and grant number MOST 110-2221-E-507-002 to J.-Y.W.

**Data Availability Statement:** Not applicable.

**Conflicts of Interest:** The authors declare no conflict of interest.

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
