# Peer review of "Application of Biotransformation-Guided Purification in Chinese Medicine: An Example to Produce Butin from Licorice"

_catalysts, doi:10.3390/catal12070718_

Round 1

Reviewer 1 Report

The authors proposed to evaluate the hydroxylation capacity of the BmTYR on Licorice extract.

Their results showed that this recombinant enzyme is able to biotransform liquiritigenin into butin and thus to greatly improve the antioxidant capacity of the extract.

The present work is of interest, but I regret the authors did not discuss the comparison of the present method of obtaining butin as compared to the other natural sources of this antioxidant flavonoid. What is the interest of using biotransform licorice extract as compared to these natural sources?

Author Response

Response:

Thank you for the comment. Some studies reported that butin could be isolated from natural sources [15, 22-23]. The typical purification process of natural products includes (1) crude extraction, (2) partition, and (3) repeated chromatography with different resins. The BGP process introduces an enzymatic biotransformation process into the abovementioned repeated chromatography to isolate bioactive compounds. Thus, one can isolate the biotransformation products and ignore the other unreacted compounds although they can be further purified later. Taking butin as an example, in a typical purification process, 55 mg butin could be purified from 1 kg seeds of Vernonia anthelmintica with recovery yield (0.0055%) [15]. 30 mg butin could be purified from 12 kg dried vine stems of Spatholobus suberectus with yield (0.00025%) [22], and 12 mg butin could be purified from 11.5 kg roots of Caragana pruinosa with yield (0.0001%) [23]. In contrast, through BGP process, 15 mg butin could be purified from 10 g licorice medicine with higher yield (0.15%), which are 27-, 1500-, and 600-folds higher than those from seeds of Vernonia anthelmintica, stems of Spatholobus suberectus, and roots of Caragana pruinosa, respectively. Thus, one of the largest advantages of the BGP process is the ability to purify higher yields of bioactive compound (butin) biotransformed from natural sources (licorice medicine), especially when the natural sources contain high amount of biotransformation precursors (liquiritigenin within licorice medicine) (Figure 1). In addition, a good biotransformation enzyme, such as BmTYR [5-6], is another key issue for better conversion rate in the BGP process. To make the readers clearer, we have added the description above at the end of section 2.4 at page 8 of the revised manuscript.

Reviewer 2 Report

I read carefully the work presented to me for review, entitled Application of Biotransformation-Guided Purification in Chinese Medicine: An Example to Produce Butin from Licorice. The work concerns an interesting aspect concerning the enzymatic synthesis of Butin from Licorice. The experiments are planned and performed correctly. And the conclusions drawn are supported by appropriate experimental data. The work deserves publication in its current form

Author Response

Response: Thank you very much for the positive comment and supporting our manuscript for publication.
